# WEIGHT UNCERTAINTY IN INDIVIDUAL TREATMENT EFFECT

## ABSTRACT

The estimation of individual treatment effects (ITE) has recently gained significant attention from both the research and industrial communities due to its potential applications in various fields such as healthcare, economics, and education. However, the sparsity of observational data often leads to a lack of robustness and over-fitting in most existing methods. To address this issue, this paper investigates the benefits of incorporating uncertainty modeling in the process of optimizing parameters for robust ITE estimation. Specifically, we derive an informative generalization bound that connects to Bayesian inference and propose a variational bound in closed form to learn a probability distribution on the weights of a hypothesis and representation function. To the best of our knowledge, this is the first work on weight uncertainty for ITE estimation. Through experiments on one synthetic dataset and two benchmark datasets, we demonstrate the effectiveness of our proposed model in comparison to state-of-the-art methods. Moreover, we conduct experiments on a real-world dataset in recommender scenarios to verify the benefits of uncertainty in causal inference. The results of our experiments provide evidence of the practicality of our model, which aligns with our initial expectations. To facilitate this research direction, we release our project at https://github-uite.github.io/uite/.

## 1 INTRODUCTION

Recently, Individual Treatment Effect (ITE) has been widely used on a large amount of real-world applications Wang et al. (2021; 2022). Basically, the fundamental goal of ITE estimation lies in the accurate understanding of the causal effect of actions for given a unit's covariates, and further providing suitable actions to the unit. For example, a doctor wants to know which medication will result in a better outcomes for a patient; or a job training institution faces the challenge of determining which training program can provide the maximum benefit to a job seeker. Nevertheless, accurate and correct estimation of ITE is often a challenging task due to the lack of counterfactuals. To this end, people have designed a lot of promising models. For example, re-weighting methods Austin (2011); Imai & Ratkovic (2014); Fong et al. (2018) utilize an Inverse Propensity Weighting (IPW) mechanism to alleviate the serious selection bias in terms of the covariates; traditional machine learning based models Breiman (2001); Wager & Athey (2018); Hill (2011), like Random Forests (RF), Causal Forests (CF), etc, use more flexible structure to model the distribution of counterfactuals; and recently more advanced technique, like Integral Probability Metric (IPM), are applied to learn the invariant representation in latent space of unit's feature and accordingly to measure the ITE based on two separated hidden vector Shalit et al. (2017); Johansson et al. (2016); Qin et al. (2021).

While the above models have achieved remarkable successes, they mostly train their models based on the observational datasets, which are usually extremely sparse in terms of the unit-action interactions in observational data. In clinical datasets, for example, only a bit small part of the population is engaged with medication and each patient may only interact with several corresponding treatments. Such sparse unit-action interactions may only reveal partial principles of causal effects, and the models learned based on them can be sensitive to the observed samples and cause critical over-fitting problems. In addition to that, since the lack of sufficient samples to characterize the robust requirements, the causal inference models may have difficulties to predict reliable results , which is crucial to the estimation of ITE.

To alleviate this problem, in this paper, we propose to measure the **U**ncertainty in **I**ndividual **T**reatment **E**ffect (called UITE for short) estimation By Bayesian inference. In general, we focus on investigating the benefits of modeling uncertainty in the process of optimizing parameters. In specific, we first derive an informative generalization bound based on the previous works Shalit et al. (2017). The bound are related to Bayesian Inference, which consists of one of the most important term, KL divergence distance between the prior and posterior distribution of weights in hypothesis $f$ and representation function $\Phi$. According to that above bound, we adopt the advanced variational approximation to calculate the posterior distribution of the causal effects, which help to improve the robustness of our model. In essence, the process of inferring predictions for unseen data through Bayesian inference can be equated to testing in the context of ensemble learning. This approach has been demonstrated to effectively address the issue of over-fitting in machine learning models.

In our experimental study, we have evaluated the performance of our proposed model against the current state-of-the-art methods. The comparison was made using one synthetic dataset and two benchmark datasets, which allowed us to empirically demonstrate the effectiveness of our model. Additionally, we also conducted experiments on a real-world dataset in the context of recommender systems to validate the advantages of incorporating uncertainty in causal inference. The results obtained from these experiments provide strong evidence of the practicality of our model, which is consistent with our initial expectations.

The main contributions of this paper can be summarized as follows: (1) To achieve an accurate estimation of ITE, we propose to build a robust casual model via Bayesian Inference. To the best of our knowledge, this is the first work on weight uncertainty for ITE estimation; (2) We theoretically analyze the loss function of ITE and derive an informative generalization bound that connects to Bayesian inference; (3) Based on the obtained bound, we propose a variational bound in closed form to learn the uncertainties of ITE; (4) We conduct extensive experiments to demonstrate the effectiveness of our framework based on real-world and benchmark datasets.

## 2 PROBLEM FORMULATION

In this paper, we adopt the Neyman-Rubin potential outcomes framework Rubin (2005) to formulate our problems. Specifically, we use $x \in \mathcal{X}$ to denote the features or covariates of a unit, and $t \in \mathcal{T}$ to represent a treatment or intervention on a unit. Throughout this paper, we focus on the binary treatment case, where $\mathcal{T} = 0, 1$ and $y \in \mathcal{Y}$ represents the factual outcome. In practice, we can only observe the factual outcome with respect to treatment assignment, i.e., $y = Y_0$ if $t = 0$, and $y = Y_1$ otherwise, where $Y_t$ denotes the potential outcome for treatment $t$. $\tau(x)$ is the the Individual Treatment Effect (ITE) on a unit $x$, or also known as the conditional average treatment effect (CATE) Shalit et al. (2017):

$$\tau(x) := \mathbb{E}[Y_1 - Y_0 | x] \tag{1}$$

The fundamental problem of causal inference is that for any unit $x$ in our setting, we can only observe either $Y_1$ or $Y_0$, but never both. In this paper, we follow recent works Qin et al. (2021); Wager & Athey (2018) and implicitly assume that there is some observable data. Formally, let $D = (x_i, t_i, y_i)_{i=1}^m$ denote the training data drawn from the observational data distribution $\mathcal{D}$, i.e., $D \in \mathcal{D}$. To ensure that the potential outcomes are identifiable from factual observational data, we require the three significant assumptions of **Consistency, Ignorability**, and **Positivity** Yao et al. (2021).

**Assumption 1.** *Consistency. If $T = a$ for a given unit, then the potential outcome for treatment $a$ is the same as the observed outcome: $Y_t = y$*

**Assumption 2.** *Positivity. For any unit covariates $X$, if $p(X) \neq 0$, then:*

$$p(T = t | X = x) > 0, \forall\, t \text{ and } x \tag{2}$$

**Assumption 3.** *Ignorability. For a given unit, the treatment is independent of the potential outcomes if given the unit covariates $X$:*

$$T \perp Y_1, Y_0 | X \tag{3}$$

Based on these assumptions, we can formulate the problem of estimating ITE as $\tau(x) = \mathbb{E}[Y|x, t = 1] - \mathbb{E}[Y|x, t = 0]$, which only involves statistical quantities that can be derived from observational data.

**Definition 1.** Let $\Phi : \mathcal{X} \to \mathcal{R}$ be a representation function, $f : \mathcal{R} \times \{0,1\} \to \mathcal{Y}$ be a hypothesis predicting the outcome of a unit's features $x$ given the representation covariates $\Phi(x)$ and the treatment assigment $t$. Let $L : \mathcal{Y} \times \mathcal{Y} \to \mathbb{R}_+$ be a loss function. The expected factual and counterfactual losses of $\Phi$ and $f$ are:

$$\epsilon_F(f, \Phi) = \int_{\mathcal{X} \times \mathcal{T} \times \mathcal{Y}} L(y, f(\Phi(x), t)) p(x, t, y) dx dt dy$$

$$\epsilon_{CF}(f, \Phi) = \int_{\mathcal{X} \times \mathcal{T} \times \mathcal{Y}} L(y, f(\Phi(x), 1 - t)) p(x, 1 - t, y) dx dt dy$$

(4)

It is evident that $\epsilon_F$ measures the accuracy of $f$ and $\Phi$ in predicting the factual outcomes based on unit features and treatments sampled from the same distribution as our data sample. On the other hand, $\epsilon_{CF}$ aims to measure the accuracy of predicting the counterfactual outcomes based on the same unit features but with the opposite treatment.

**Definition 2.** The expected factual treated and control losses are:

$$\epsilon_F^{t=1}(f, \Phi) = \int_{\mathcal{X} \times \mathcal{Y}} L(y, f(\Phi(x), 1)) p(x, y | T = 1) dx dy$$

$$\epsilon_F^{t=0}(f, \Phi) = \int_{\mathcal{X} \times \mathcal{Y}} L(y, f(\Phi(x), 0)) p(x, y | T = 0) dx dy$$

(5)

Accordingly, we can obtain an immediate results $\epsilon_F(f, \Phi) = p(t = 1)\epsilon_F^{t=1}(f, \Phi) + p(t = 0)\epsilon_F^{t=0}(f, \Phi)$.

**Definition 3.** The estimation of treatment effect by an hypothesis $f$ and a representation function $\Phi$ for unit $x$ is:

$$\hat{\tau}(x) = f(\Phi(x), 1) - f(\Phi(x), 0)$$

(6)

**Definition 4.** The expected Precision in Estimation of Heterogeneous Effect (PEHE) Hill (2011) loss of $f$ and $\Phi$ is:

$$\epsilon_{PEHE}(f) = \int_{\mathcal{X}} (\hat{\tau}(x) - \tau(x))^2 p(x) dx$$

(7)

**Definition 5.** Integral Probability Metric (IPM). For two probability density functions $p, q$ defined over $\mathcal{S} \in \mathbb{R}^d$, and for a function family $G$ of functions $g : \mathcal{S} \to \mathbb{R}$, The IPM is Shalit et al. (2017):

$$IPM_G(p, q) := \sup_{g \in G} \left| \int_{\mathcal{S}} g(s)(p(s) - q(s)) \right|$$

(8)

From the definition, it is evident that IPM measures the distance between two distributions. For rich enough function families $G$, IPM is a true metric over the corresponding set of probabilities Shalit et al. (2017); Qin et al. (2021). When we let $G$ satisfy the family of 1-Lipschitz functions, i.e., $G = g : ||g||p \le 1$, we obtain the Wasserstein distance denoted by $Wass_G(\cdot, \cdot)$. On the other hand, when $G = g \in \mathcal{H}$ s.t. $||g||\mathcal{H} \le 1$, we derive Maximum Mean Discrepancy (MMD) denoted by $MMD_G(\cdot, \cdot)$, where $\mathcal{H}$ represents a reproducing kernel Hilbert space (RKHS) Sriperumbudur et al. (2009). In the rest of the paper, we consider an estimation for ITE in the form of $f(\Phi(x), 1) - f(\Phi(x), 0)$ using the two types of distribution metrics.

## 3   BAYESIAN INFERENCE FOR ITE

In this section, we elaborate on the variational bound that are leveraged to closer posterior distribution of the weights. The weights are used to estimate the ITE. In essence, Bayesian inference aims to calculate the posterior distribution of the weights given the observational training data and then use this distribution to infer the predictions of unseen data by taking expectations. Specifically, we have $p(\hat{Y} | \mathcal{D}) = \mathbb{E}_{p(\boldsymbol{w} | D)}[p(\hat{y} | x, t)]$, in which $p(\boldsymbol{w} | D)$ is the posterior distribution. With the Bayes rules, we have,

$$p(\boldsymbol{w} | D) = \frac{p(\boldsymbol{w}) p(D | \boldsymbol{w})}{\int p(D) dD}$$

(9)

Unfortunately, the term $\int p(D) dD$ is intractable for neural networks of any practical size Blundell et al. (2015). Instead, we adopt the advanced variational approximation to close the posterior

distribution. That is

$$\mathrm{KL}[q(\boldsymbol{w}|\boldsymbol{\theta})||p(\boldsymbol{w}|D)] = \underbrace{\log(p(D))}_{Constant} - \underbrace{\int q(\boldsymbol{w}|\boldsymbol{\theta}) \log \frac{p(\boldsymbol{w})p(D|\boldsymbol{w})}{q(\boldsymbol{w}|\boldsymbol{\theta})}}_{VLB} \tag{10}$$

The constant term $\log(p(D))$ does not affect the optimization objective. Thus, we instead maximize its variational lower bound (VLB):

$$\begin{aligned} L_{VLB} &= \int q(\boldsymbol{w}|\boldsymbol{\theta}) \log \frac{p(\boldsymbol{w})p(D|\boldsymbol{w})}{q(\boldsymbol{w}|\boldsymbol{\theta})} \\ &= \underbrace{\mathrm{KL}[q(\boldsymbol{w}|\boldsymbol{\theta})||p(\boldsymbol{w})]}_{A} - \underbrace{\mathbb{E}_{q(\boldsymbol{w}|\boldsymbol{\theta})}[\log p(D|\boldsymbol{w})]}_{B} \end{aligned} \tag{11}$$

where A is the KL divergence between a learnable posterior distribution $q(\boldsymbol{w}|\boldsymbol{\theta})$ and a prior distribution $p(\boldsymbol{w})$ and B refer to as the likelihood function on current datasets. Naively minimizing this VLB is computationally prohibitive Blundell et al. (2015). Therefore, various approximations and optimization techniques, such as gradient descent, are used to efficiently estimate the posterior distribution of the weights.

## 4 THEORETICAL INSIGHTS

To understand the uncertainty of ITE estimation more in depth, we analyze it from the theoretical perspective. For clear presentation, we first introduce the error bounds in estimating the ITE, and then shed light on the obtained theoretical results. We present the proofs in Appendix.

**Proposition 1.** *Shalit et al. (2017). Let $\Phi : \mathcal{X} \to \mathcal{R}$ be a one-to-one representation function and $f : \mathcal{R} \times \mathcal{T} \to \mathcal{Y}$ be a hypothesis. Let $G$ be a family of functions $g : \mathcal{R} \to \mathcal{Y}$. Assume that there exists a $\ell_2$ loss, $L : \mathcal{Y} \times \mathcal{Y} \to \mathcal{R}_+$, and a constant $C_\Phi > 0$, such that for fixed $t \in \{0, 1\}$, the per-unit expected loss function $\ell_{f,\Phi}(x,t) = \int_{\mathcal{Y}} L(Y_t, f(\Phi(x),t))p(Y_t|x)dY_t$ obey $\frac{1}{C_\Phi} \cdot \ell_{f,\Phi}(x,t) \in G$. Then,*

$$\epsilon_{PEHE}(f, \Phi) \le 2(\epsilon_{CF}(f, \Phi) + \epsilon_F(f, \Phi) - C_Y)$$
$$\le 2\left(\epsilon_F^{t=0}(f, \Phi) + \epsilon_F^{t=1}(f, \Phi)\right) + 2\left(C_\Phi \cdot IPM_G(p_\Phi^{t=1}, p_\Phi^{t=0}) - C_Y\right)$$

*where $p_\Phi^{t=1} = p(\Phi(x)|t=1)$, $p_\Phi^{t=0} = p(\Phi(x)|t=0)$ are the treated and control distributions define over $\mathcal{R}$ separately, and $C_Y$ is a constant induced over the variance of the outcomes $Y_t$.*

*Remark.* Proposition 1 provides an upper bound for the expected ITE estimation error, which is bounded by the sum of the standard regression generalization error and the IPM distance between the treated and control distributions. With the bound in mind, some exiting methods try to minimize the generalization bound using a neural network Qin et al. (2021). Generally speaking, they proceed in two steps: (1) minimizing the empirical losses for accurate estimation of ITE; (2) optimizing distribution discrepancy IPM for learning invariant representation that are used to facilitate the predictions of ITE. However, since the sparsity of the observational datasets, most exiting casual methods are not capable of building a robust model. To this end, we propose to calculate the ITE via Bayesian inference. By incorporating the uncertainty into the causal inference, we have the following theory:

**Theorem 1.** *Given a distribution $\mathcal{D}$ over $\mathcal{X} \times \mathcal{T} \times \mathcal{Y}$. Let $\Phi : \mathcal{X} \to \mathcal{R}$ be a one-to-one representation function and $f : \mathcal{R} \times \mathcal{T} \to \mathcal{Y}$ be a hypothesis. Let $\mathcal{F}$ be a set of hypothesis $f$, $\pi$ and $\rho$ be a prior and a posterior distributions over $\mathcal{F}$, respectively. Assume that there exists a $\ell_2$ loss, $L : \mathcal{Y} \times \mathcal{Y} \to \mathcal{R}_+$, any $\delta \in (0, 1]$, with probability at least $1 - \delta$ over the choice of $D \in \mathcal{D}$, we have*

$$\forall \rho \text{ on } \mathcal{F} :$$

$$\mathbb{E}_{f,\Phi\sim\rho} \epsilon_F(f, \Phi) \le \mathbb{E}_{f,\Phi\sim\rho} \frac{1}{m} \sum_{i=1}^m L(y_i, f(\Phi(x_i), t_i)) + \frac{1}{m}\left[KL(\rho|\pi) + \ln\frac{1}{\delta} + \Psi(m)\right],$$

$$where \quad \Psi(m) = \ln \mathbb{E}_{f,\Phi\sim\pi} \mathbb{E}_{D'\sim\mathcal{D}} \exp\left[m\left(\epsilon_F(f, \Phi) - \frac{1}{m}\sum_{i=1}^m L(y_i', f(\Phi(x_i)', t_i'))\right)\right]$$

*Remark.* Theorem1 provide a PAC-Bayes bound for the expected factual losses of $\Phi$ and $f$, which brings probably approximately correct generalization bounds on the quantity $\mathbb{E}_{f,\Phi\sim\rho} \epsilon_F(f, \Phi)$ for the

given empirical estimate and some other parameters. Among these, our derived bound highly rely on the Kullback-Leibler divergence $KL(\rho|\pi) = \mathbb{E}_{f,\Phi \sim \rho} \ln[\rho(f,\Phi)/\pi(f,\Phi)]$ between the prior and posterior distribution. By integrating this results with the proposition 1, we have the following theory:

**Theorem 2.** *Under the conditions of Definition 1, Proposition 1 and Theorem 1 with probability at least $1 - \delta$,*

$$\forall \rho \text{ on } \mathcal{F}:$$

$$\mathbb{E}_{f,\Phi \sim \rho} \epsilon_{PEHE}(f,\Phi) \leq 4 \left( \mathbb{E}_{f,\Phi \sim \rho} \frac{1}{m} \sum_{i=1}^{m} L(y_i, f(\Phi(x_i), t_i)) + \frac{1}{m} \left[ KL(\rho|\pi) + \ln \frac{1}{\delta} + \Psi(m) \right] \right)$$
$$+ 2 \left( C_\Phi \cdot IPM_G(p_\Phi^{t=1}, p_\Phi^{t=0}) - C_Y \right)$$

*Remark.* Theorem 2 provides an informative generalization bound for estimating the uncertainty of ITE. This bound is mainly composed of the empirical regression losses on the training set, as well as the KL divergence and IPM distance between the treated and control groups. With the exception of the distance metric constant term, all of these quantities can be empirically estimated or approximated using deep learning techniques. The upper bound decreases as these three quantities become smaller. Therefore, Theorem 2 instructs us to measure ITE from the perspective of Bayesian inference, which can help to build a robust ITE model and alleviate over-fitting problems.

## 5 THE PROPOSED METHOD

Based on above theoretical analysis, we propose a method called UITE (**U**ncertainty in **I**ndividual **T**reatment **E**ffect), which aims to calculate the uncertainty of ITE. By taking expectations among posterior distribution, we can write the following objective:

$$\mathbb{E}_{f,\Phi \sim \rho} \left[ \sum_{i=1}^{m} \frac{\gamma_i}{m} \cdot L(y_i, f(\Phi(x_i), t_i)) + \log \frac{\pi}{\rho} + \alpha \cdot \text{IPM}_G(\hat{p}_\Phi^{t=1}, \hat{p}_\phi^{t=0}) \right] \tag{12}$$

where the weights $\gamma_i$ compensates for the difference in treatment group size Shalit et al. (2017). it often be calculated by the proportion of treated units in the population. $\hat{p}_\Phi^{t=1}$ and $\hat{p}_\Phi^{t=0}$ are learned high-dimensional representation for treated and control groups respectively. Note that $\text{IPM}_G(\cdot, \cdot)$ is the distance metric and the specific implementations of it depends on the selected distance metric. $\log \frac{\pi}{\rho}$ is the KL divergence term, which aims to trade-off the prior and learnable posterior distribution of weights. Careful readers may find that the term $\Psi(m)$ has been discarded in our objective. In fact, the term $\Psi(m)$ is determined by the empirical loss $L$. We refer to the model minimizing equation 12 with WASS distance metric as UITE$_{WASS}$ and the variant with MMD distance metric as UITE$_{MMD}$. Both models are trained by the adaptive moment estimation (Adam) Kingma & Ba (2014). The details are described in Appendix.

## 6 RELATED WORK

**Estimating Individual Treatment Effect.** How to effectively and correctly measure individual treatment effect has recently attracted increasing attention from the research community. It basically aims to discover the underlying patterns of the distribution between treated and control group. To achieve this goal, the widely used techniques for individual treatment effect inference are traditional machine learning, including Random Forests (RF) Breiman (2001), Causal Forests (CF) Wager & Athey (2018), etc. These methods have more flexibility and predictive ability in balancing the distribution between treated and control groups. In addition, some studies resort to adapting more sophisticated mechanisms to causal effect inference, and in particular to measure individual level treatment effect. For example, Causal Effect Variational Autoencoder (CEVAE) Louizos et al. (2017) leverage Variational Autoencoders to obtain the unobserved confounders and simultaneously infer causal effects, DragonNet Shi et al. (2019) design three-head components to predict the treatment effects as well as adjust the distribution by a process of inferring treatments. Besides, more cutting-edge mechanism like Integral Probability Metric (IPM) Qin et al. (2021); Johansson et al. (2016) are applied to minimize generalization bound for ITE estimation, which is composed of factual loss and the discrepancy between the treated and control distributions. The representative CFR Shalit et al. (2017) method enforce the similarity between the distributions of treated and control groups in the representation space by a penalty term IPM. While the boundary of estimation of ITE estimation from

observational data has been pushed by these models, an important problem is still under-explored, that is the issues of over-fitting when the input observational data is extremely sparsity. In this paper, we bridge this gap by incorporating the Bayesian inference into the ITE estimation, which will help to enhance the model robustness as well as improving its performance.

**Bayesian Inference.** Bayesian inference has been widely investigated in the machine learning community. In general, The keystone of Bayesian inference is to study an averaging of losses according to a posterior distribution of weights in a neural network Germain et al. (2016); Antorán et al. (2020), and then predict the unseen test data in a way of ensemble learning. As we know, there are two types of definitions on the expectations estimation. For the first one, it aims to derive an unbiased estimation by Monte Carlo sampling Germain et al. (2016); Maddox et al. (2019). For the second one, it suggests to find a variational approximation to the Bayesian posterior distribution on the weights Daxberger et al. (2021); Immer et al. (2021). In addition, some inspiring and insightful theoretical results are proposed to guarantee that the expected test error are under control Germain et al. (2016; 2009). Motivated by these promising works, in this paper, we propose to build a robust casual model by using the expectations estimation on Bayesian inference.

# 7 BENCHMARK EXPERIMENTS

## 7.1 EXPERIMENT SETUP

The estimation of Individual Treatment Effects (ITE) is comparatively more challenging than other machine learning tasks. This is primarily because in real-world scenarios, the ground-truth treatment effect is rarely available. To evaluate the efficacy of the proposed framework, we performed experiments on one synthetic example, named **Sim**, and two widely recognized benchmark datasets, namely **ACIC** and **IHDP**. The more details about them are presented in the following: **ACIC 2016** is a common benchmark dataset introduced by Dorie et al. (2019), which was developed for the 2016 Atlantic Causal Inference Conference competition data Dorie et al. (2019). It comprises 4,802 units (28% treated, 72% control) and 82 covariates measuring aspects of the linked birth and infant death data (LBIDD). The dataset are generated randomly according to the data generating process setting. We conduct experiments over randomly picked 100 realizations with 63/27/10 train/validation/test splits. **IHDP** Hill (2011) introduced a semi-synthetic dataset for causal effect estimation. The dataset was based on the Infant Health and Development Program (IHDP), in which the covariates were generated by a randomized experiment investigating the effect of home visits by specialists on future cognitive scores. it consists of 747 units(19% treated, 81% control ) and 25 covariates measuring the children and their mothers. Following the common settings in Qin et al. (2021); Shalit et al. (2017), We average over 1000 replications of the outcomes with 63/27/10 train/validation/test splits. **Data Simulation.** To verify the effectiveness of our framework on unbiased data, we adopt the generation process proposed in Assaad et al. (2021); Louizos et al. (2017) to simulate the treatment effect as:

$$
\begin{aligned}
&\mathrm{x}_i \sim \mathcal{N}(\mu_X, \sigma_X^2); \ \ \mathrm{t}_i|\mathrm{x}_i \sim \mathrm{Bernoulli}(\sigma(\mathrm{x}_i^T \beta_T)) \\
&\epsilon_i \sim \mathcal{N}(0, \sigma_Y^2); \ \ \mathrm{y}_i(0) = \mathrm{x}_i^T \beta_0 + \epsilon_i \\
&\mathrm{y}_i(1) = \mathrm{x}_i^T \beta_0 + \mathrm{x}_i^T \beta_1 + \theta + \epsilon_i
\end{aligned}
\tag{13}
$$

where $\sigma$ is the logistic sigmoid function. This generation process satisfies the assumptions of ignorability and positivity. We randomly construct 100 replications of such datasets with 10,000 units (50% treated, 50% control) and 50 covariates by setting $\sigma_X$ and $\sigma_Y$ both to 0.5, $\beta_T$, $\beta_0$ and $\beta_1$ are sampled from $\mathcal{N}(0,1)$. The statistics of these datasets are summarized in Appendix.

We compare our model with the following 11 representative baselines: Random Forests (RF) Breiman (2001), Causal Forests (CF) Wager & Athey (2018), Causal Effect Variational Autoencoder (CEVAE) Louizos et al. (2017), DragonNet Shi et al. (2019), Meta-Learner algorithms S-Learner Nie & Wager (2021) and T-Learner Künzel et al. (2019), Balancing Neural Network (BNN) Johansson et al. (2016), Treatment-Agnostic Representation Network (TARNet) Shalit et al. (2017) as well as CounterFactual Regression with the Wasserstein metric (CFR$_{WASS}$) Shalit et al. (2017) and the squared linear MMD metric (CFR$_{MMD}$) Shalit et al. (2017), along with a extension of CRF method Query-based Heterogeneous Treatment Effect estimation (QHTE) Qin et al. (2021). A detailed description of the implementations and datasets used in this study can be found in the Appendix. For further information regarding the implementation of all adopted baselines, our methods, and full experimental settings, please refer to Appendix.

Table 1: Individual treatment effect estimation on ACIC, IHDP and Sim test set. The top module consists of baselines from recent works. The bottom module consists of our proposed methods. In each module, we present each of the result with form mean ± standard deviation and we use bold fonts to label the best performance. Lower is better.

| Datasets | ACIC | | IHDP | | Sim | |
|---|---|---|---|---|---|---|
| Metric | $\sqrt{\epsilon_{PEHE}}$ | $\epsilon_{ATE}$ | $\sqrt{\epsilon_{PEHE}}$ | $\epsilon_{ATE}$ | $\sqrt{\epsilon_{PEHE}}$ | $\epsilon_{ATE}$ |
| RF | 3.09 ± 1.48 | 1.16 ± 1.40 | 4.61 ± 6.56 | 0.70 ± 1.50 | 3.36 ± 0.01 | 3.87 ± 0.01 |
| CF | 1.86 ± 0.73 | 0.28 ± 0.27 | 4.46 ± 6.53 | 0.81 ± 1.36 | 1.81 ± 0.04 | 0.08 ± 0.06 |
| S-learner | 3.86 ± 1.45 | 0.41 ± 0.35 | 5.76 ± 8.11 | 0.96 ± 1.8 | 1.92 ± 0.05 | 0.06 ± 0.05 |
| T-learner | 2.33 ± 0.86 | 0.79 ± 0.68 | 4.38 ± 7.85 | 2.16 ± 6.17 | 0.57 ± 0.02 | 0.03 ± 0.02 |
| CEVAE | 5.63 ± 1.58 | 3.96 ± 1.37 | 7.87 ± 7.41 | 4.39 ± 1.63 | 1.92 ± 0.05 | 0.12 ± 0.18 |
| BNN | 2.00 ± 0.86 | 0.43 ± 0.36 | 3.17 ± 3.72 | 1.14 ± 1.70 | 1.08 ± 0.09 | 0.26 ± 0.15 |
| DragonNet | 1.26 ± 0.32 | 0.15 ± 0.13 | 1.46 ± 1.52 | 0.28 ± 0.35 | 0.43 ± 0.05 | 0.09 ± 0.07 |
| TARNet | 1.30 ± 0.46 | **0.15 ± 0.12** | 1.49 ± 1.56 | 0.29 ± 0.40 | 0.45 ± 0.04 | 0.09 ± 0.06 |
| CFR$_{MMD}$ | **1.24 ± 0.31** | 0.17 ± 0.14 | 1.51 ± 1.66 | 0.30 ± 0.52 | 0.46 ± 0.04 | 0.09 ± 0.07 |
| CFR$_{WASS}$ | 1.27 ± 0.38 | 0.15 ± 0.12 | 1.43 ± 1.61 | 0.27 ± 0.41 | 0.49 ± 0.05 | 0.10 ± 0.07 |
| QHTE | 1.32 ± 0.41 | 0.19 ± 0.18 | 1.83 ± 1.90 | 0.34 ± 0.43 | 0.51 ± 0.06 | 0.18 ± 0.06 |
| UITE$_{MMD}$ | 2.08 ± 0.03 | 0.16 ± 0.12 | 0.84 ± 0.06 | 0.19 ± 0.13 | **0.13 ± 0.01** | **0.04 ± 0.02** |
| UITE$_{WASS}$ | 2.89 ± 0.99 | 0.28 ± 0.22 | **0.75 ± 0.05** | **0.16 ± 0.10** | 0.17 ± 0.01 | 0.04 ± 0.03 |

## 7.2 IMPLEMENTATION DETAILS

We have implemented our methods based on CFR Shalit et al. (2017). For UITE, we have used the same set of hyperparameters across all three datasets. Specifically, we have adopted 3 fully-connected exponential-linear layers for the representation function $\Phi$ and 3 similar architecture layers for the ITE prediction function $f$. The only difference is that the layer sizes are 200 for the former and 100 for the latter. To facilitate training, batch normalization Ioffe & Szegedy (2015) has been applied, and ReLU (Rectified Linear Unit) Agarap (2018) has been used as the activation function for all layers except the output layer. In the main optimization objective, we have set $\alpha$ to 1, the prior distribution of weights $p(\boldsymbol{w})$ to a standard normal distribution $\mathcal{N}(0,1)$, and the posterior distribution of weights $q(\boldsymbol{w}|\boldsymbol{\theta})$ to a Gaussian distribution $\mathcal{N}(\mu, \sigma^2)$ with learnable mean $\mu$ and variance $\sigma$. For further details on the implementation of all adopted baselines and our methods, as well as full experimental settings, please refer to Appendix. As mentioned in section 4, we use the Wasserstein (UITE$_{WASS}$) and squared linear MMD (UITE$_{MMD}$) distances to penalize imbalance. We evaluate the quality of individual treatment effects using commonly used metrics, such as Rooted Precision in Estimation of Heterogeneous Effect (PEHE) Hill (2011) and Mean Absolute Error (ATE) Shalit et al. (2017). Formally, they are defined as:

$$\sqrt{\epsilon_{PEHE}} = \sqrt{\frac{1}{n}\sum_{i=1}^{n}(\hat{\tau}_i - \tau_i)^2}, \epsilon_{ATE} = |\frac{1}{n}\sum_{i=1}^{n}(\hat{\tau}) - \frac{1}{n}\sum_{i=1}^{n}(\tau)| \tag{14}$$

where $\hat{\tau}_i$ and $\tau_i$ stand for the predicted ITE and the ground truth ITE for the $i$-th instance respectively.

## 7.3 RESULTS

The overall comparison results are presented in Table 1, from which we can see: among the baselines, IPM-based methods, such as CFR$_{MMD}$ and CFR$_{WASS}$, can usually achieve better performance than the ones like RF and S-Learner who are based on traditional machine learning. This observation is consistent to the previous work that using distance metric to balance the distribution between the treated and control groups is indeed an effective way for learning the invariant representations in latent space. And the invariant representations is crucial to the independent conditions of treatment $t$ and potential outcome $Y_t$ for given unit covariates. Encouragingly, our model can achieve the best performance in most cases, and the results are consistent across different datasets and evaluation metrics. This observation demonstrates the effectiveness and generality of our model. We speculate that due to the averaging results of ITE estimation in terms of a posterior distribution of weights in the testing sets may help to improve the model robustness and make our framework more tolerate, and

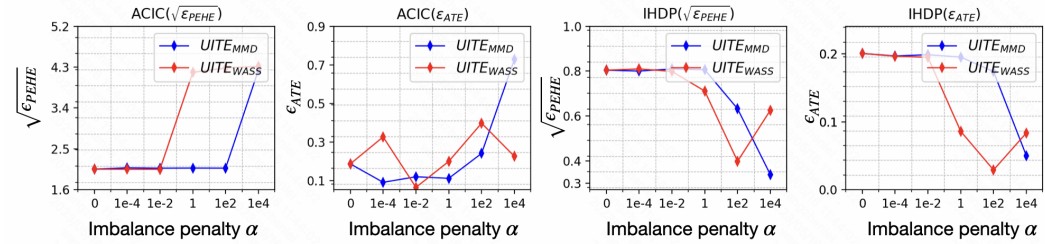

Figure 1: Influence of the regularization penalty $\alpha$ on our model performance in terms of $\sqrt{\epsilon_{PEHE}}$ and $\epsilon_{ATE}$. The performances of different distance metric implementations are labeled with different colors. Lower is better.

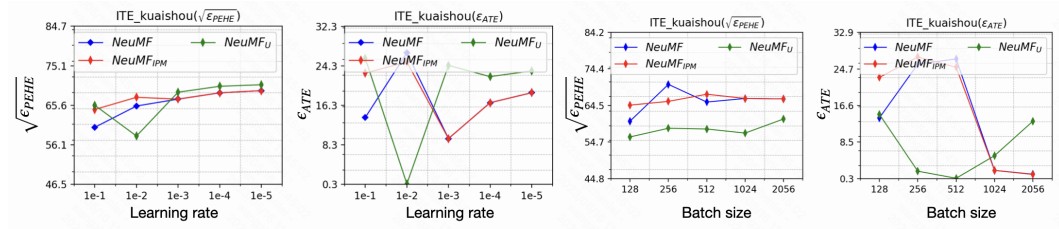

Figure 2: Performance comparison between recommender base model NeuMF and its variants on casual inference NeuMF$_{IPM}$ and NeuMF$_U$. The performances of different types of recommender models are labeled with different colors. Lower is better.

thus lead to better performances. The results aligns with our expectations that uncertainty estimation of ITE can make our model more accurate and predictable.

## 7.4 UNCERTAINTY STUDY

In this section, we investigate the model's uncertainty from the perspective of robustness. As we mentioned above, estimating the uncertainty of ITE can help to improve the model's robustness and alleviate the issues of over-fitting. To verify that, we conduct experiments with the robustness certification of model. To do that, we compare our methods to the representative deep learning methods BNN, DragonNet, TARNet, CFR$_{WASS}$ and CFR$_{MMD}$ on datasets of ACIC and IHDP. For a test unit covariate $x$, we add noise in $\mathbb{U}(-1, 1)$ to it, and then generate a new one $x'$ to substitute the original unit covariates. The results are presented in Table 2, from which we can see: By introducing small perturbations to the unit covariates, we observed a joint degradation in the performance of all models across the ACIC and IHDP datasets compared to Table 1. However, it is inspiring to note that our framework still outperform the base models in most cases. This observation demonstrate that our framework can improve model robustness even when the unit covariates are perturbed.

## 7.5 PARAMETER STUDY

As described in the preceding sections, our main optimization objective consists of several terms, including the imbalance penalty $\alpha$, which determines the strength of balancing the distribution between the treated and control groups. Readers may be interested in how this term contributes to the final performance. To address this question and illustrate the influence of this term, we conduct parameter studies in this section. The hyperparameter settings follow the above experiments, and we compare our model by varying the imbalance penalty $\alpha$. For the optimization objective 12, the influence of the IPM decreases as the imbalance penalty $\alpha$ becomes smaller. We tune $\alpha$ in the range of [0, 1e-4, 1e-2, 1, 1e2, 1e4]. The results are presented in Figure 1. We observe that the best performance is usually achieved when $\alpha$ is moderate. This finding is consistent with our opinion in section 4, which suggests that too small of an $\alpha$ may introduce too much imbalance representation into the training process, while too large of an $\alpha$ may severely impact the predictions made by the ITE model. By tuning $\alpha$ within appropriate ranges, we can achieve better trade-offs and improve the ITE estimation performance.

Table 2: Performance comparison between the model testing in the original test sets and with small perturbation in test sets. We use "X-Noisy" to represent the test set with noisy when the model is "X". We highlight the best performance with bold fonts. Lower is better.

| Datasets | ACIC | | IHDP | |
|---|---|---|---|---|
| Metric | $\sqrt{\epsilon_{PEHE}}$ | $\epsilon_{ATe}$ | $\sqrt{\epsilon_{PEHE}}$ | $\epsilon_{ATe}$ |
| BNN-Noisy | $4.84 \pm 0.15$ | $1.50 \pm 0.46$ | $3.25 \pm 0.39$ | $3.03 \pm 0.55$ |
| DragonNet-Noisy | $1.55 \pm 0.10$ | $0.21 \pm 0.16$ | $0.65 \pm 0.08$ | $0.15 \pm 0.11$ |
| TARNet-Noisy | $\mathbf{1.48 \pm 0.10}$ | $0.22 \pm 0.17$ | $0.67 \pm 0.08$ | $0.17 \pm 0.11$ |
| CFR$_{MMD}$-Noisy | $\mathbf{1.48 \pm 0.10}$ | $0.21 \pm 0.16$ | $0.67 \pm 0.08$ | $0.16 \pm 0.12$ |
| CFR$_{WASS}$-Noisy | $1.52 \pm 0.10$ | $0.23 \pm 0.16$ | $\mathbf{0.63 \pm 0.08}$ | $0.17 \pm 0.12$ |
| UITE$_{MMD}$-Noisy | $2.11 \pm 0.04$ | $\mathbf{0.19 \pm 0.11}$ | $0.86 \pm 0.07$ | $0.19 \pm 0.13$ |
| UITE$_{WASS}$-Noisy | $3.00 \pm 0.99$ | $0.27 \pm 0.20$ | $0.76 \pm 0.05$ | $\mathbf{0.16 \pm 0.10}$ |

## 8 REAL-WORLD EXPERIMENTS

### 8.1 EXPERIMENT SETUP

To assess the practicality of our model in real-world scenarios, we collected a large dataset from Kuaishou, a popular video platform in China. The dataset consists of 120,671 users and 877,995 videos, resulting in 2,774,786 interactions, of which 86% were treated with an icon indicating entry into a live streaming session, while 14% were control interactions without an icon. Each sample in the dataset includes 8 covariates that measure various aspects of the user and video. We focus on live streaming benefits in recommender settings, and thus, we employ a promising collaborative filtering method, NeuMF He et al. (2017) , as the base model. We also design two variants of NeuMF, namely NeuMF$_{IPM}$, which utilizes IPM to reduce selection bias, and NeuMF$_U$, which leverages uncertainty with causal effect to measure the benefits of live streaming. To ensure a fair comparison, all models have the same structure and initial parameters. We present the results with hyper-parameter learning rate and batch size in range of $[0.1, 0.01, 0.001, 0.00001]$ and $[128, 256, 512, 1024]$ separately. Further details regarding the dataset and model implementations can be found in Appendix.

### 8.2 RESULTS

The results are presented in Figure 2, and we observe that for the causal effects of live streaming with a promising recommender model in real-world scenarios, NeuMF$_{IPM}$ generally outperforms NeuMF. This finding is as expected and verifies that modeling the invariant representation between treated and control groups is indeed an effective way to reduce selection bias and improve the recommender performance. Our method NeuMF$_U$ outperforms both NeuMF and NeuMF$_{IPM}$ and consistently achieves the best performance across all datasets and evaluation metrics. The reason for this improvement can be attributed to the extreme sparsity of user-item interactions in practice. By modeling the uncertainty of user behaviors, our method can provide useful signals for predicting the appearance of the current action, leading to more accurate causal effect estimation and more promising recommendation performance.

## 9 CONCLUSION

In this paper, we propose a method to build a robust causal model while mitigating the over-fitting problems that arise from the sparsity of datasets in observational studies. To achieve this goal, we investigate the benefits of modeling uncertainty in the process of optimizing parameters for ITE estimation. Specifically, we derive a loss bound in ITE that is connected to Bayesian inference. We then obtain a closed-form variational bound to learn the posterior distribution of weights, which enhances model robustness and improves performance. Our extensive experiments on both benchmark and real-world datasets demonstrate the effectiveness of our framework. However, there is still room for improvement. One can incorporate more advanced techniques to capture uncertainty for causal models and devise more suitable prior distributions for the settings. Additionally, to reduce time-consuming computations, researchers can investigate specific time steps in the process of optimizing the VLB loss.

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

# A APPENDIX

## A.1 PROOF OF THEORY 1

Motivated by the works in Germain et al. (2009; 2016), we perform our proofs in ITE now.

*Proof.* $\mathbb{E}_{f,\Phi\sim\pi} \exp\left[m\left(\epsilon_F(f,\Phi) - \frac{1}{m}\sum_{i=1}^{m} L(y_i', f(\Phi(x_i)', t_i'))\right)\right]$ is a non-negative random variable, by using Markov's inequality, we have,

$$
\begin{aligned}
\Pr_{D'\sim\mathcal{D}} &\left(\mathbb{E}_{f,\Phi\sim\pi} \exp\left[m\left(\epsilon_F(f,\Phi) - \frac{1}{m}\sum_{i=1}^{m} L(y_i', f(\Phi(x_i)', t_i'))\right)\right]\right. \\
&\left.\leq \frac{1}{\delta}\mathbb{E}_{f,\Phi\sim\pi}\mathbb{E}_{D'\sim\mathcal{D}} \exp\left[m\left(\epsilon_F(f,\Phi) - \frac{1}{m}\sum_{i=1}^{m} L(y_i', f(\Phi(x_i)', t_i'))\right)\right]\right) \geq 1-\delta
\end{aligned}
\tag{15}
$$

Then, we take the logarithm to each side of the innermost inequality and convert the $\pi$ into $\rho$ via a distribution trick, we have following results,

$$
\Pr_{D' \sim \mathcal{D}} \left( \forall \rho \, on \, \mathcal{F} : \ln \left[ \mathbb{E}_{f,\Phi \sim \rho} \frac{\pi(f,\Phi)}{\rho(f,\Phi)} \exp \left[ m \left( \epsilon_F(f,\Phi) - \frac{1}{m} \sum_{i=1}^{m} L(y_i', f(\Phi(x_i)', t_i')) \right) \right] \right] \right.
$$
$$
\left. \leq \ln \left[ \frac{1}{\delta} \mathbb{E}_{f,\Phi \sim \pi} \mathbb{E}_{D' \sim \mathcal{D}} \exp \left[ m \left( \epsilon_F(f,\Phi) - \frac{1}{m} \sum_{i=1}^{m} L(y_i', f(\Phi(x_i)', t_i')) \right) \right] \right] \right) \geq 1 - \delta \tag{16}
$$

In which, with the Jensen's inequality and the concavity of $\ln(x)$, we have,

$$
\ln \left[ \mathbb{E}_{f,\Phi \sim \rho} \frac{\pi(f,\Phi)}{\rho(f,\Phi)} \exp \left[ m \left( \epsilon_F(f,\Phi) - \frac{1}{m} \sum_{i=1}^{m} L(y_i', f(\Phi(x_i)', t_i')) \right) \right] \right]
$$
$$
\geq \mathbb{E}_{f,\Phi \sim \rho} \ln \left[ \frac{\pi(f,\Phi)}{\rho(f,\Phi)} \exp \left[ m \left( \epsilon_F(f,\Phi) - \frac{1}{m} \sum_{i=1}^{m} L(y_i', f(\Phi(x_i)', t_i')) \right) \right] \right]
$$
$$
\geq \mathbb{E}_{f,\Phi \sim \rho} \left[ \ln \frac{\pi(f,\Phi)}{\rho(f,\Phi)} + m \left( \epsilon_F(f,\Phi) - \frac{1}{m} \sum_{i=1}^{m} L(y_i', f(\Phi(x_i)', t_i')) \right) \right] \tag{17}
$$
$$
\geq \mathbb{E}_{f,\Phi \sim \rho} \ln \frac{\pi(f,\pi)}{\rho(f,\Phi)} + \mathbb{E}_{f,\Phi \sim \rho} m \left( \epsilon_F(f,\Phi) - \frac{1}{m} \sum_{i=1}^{m} L(y_i', f(\Phi(x_i)', t_i')) \right)
$$
$$
= -KL(\rho|\pi) + \mathbb{E}_{f,\Phi \sim \rho} m \left( \epsilon_F(f,\Phi) - \frac{1}{m} \sum_{i=1}^{m} L(y_i', f(\Phi(x_i)', t_i')) \right)
$$

Where $KL(\rho|\pi) = \mathbb{E}_{f,\Phi \sim \rho} \ln \frac{\rho(f,\pi)}{\pi(f,\Phi)}$.

By integrating above results into Inequality( 16) , we get,

$$
\Pr_{D' \sim \mathcal{D}} \left( \forall \rho \, on \, \mathcal{F} : \mathbb{E}_{f,\Phi \sim \rho} \left( \epsilon_F(f,\Phi) - \frac{1}{m} \sum_{i=1}^{m} L(y_i', f(\Phi(x_i)', t_i')) \right) \right.
$$
$$
\left. \leq \frac{1}{m} \left[ KL(\rho|\pi) + \ln \left[ \frac{1}{\delta} \mathbb{E}_{f,\Phi \sim \pi} \mathbb{E}_{D' \sim \mathcal{D}} \exp \left[ m \left( \epsilon_F(f,\Phi) - \frac{1}{m} \sum_{i=1}^{m} L(y_i', f(\Phi(x_i)', t_i')) \right) \right] \right] \right] \right) \geq 1 - \delta \tag{18}
$$

The proof is done. $\qquad\square$

## B  ALGORITHM AND DATASETS

The complete algorithm is presented in Algorithm 1.

The statistics of these datasets are summarized in Table 3.

---

**Algorithm 1:** Learning algorithm of our model

---

1   Indicate the observational data $(x_1, t_1, y_1), ..., (x_m, t_m, y_m)$.

2   Indicate the scaling parameter $\alpha$.

3   Initialize all the model parameters.

4   Indicate the epoch number $E$.

5   Compute $u = \frac{1}{m} \sum_{i=1}^{m} t_i$.

6   Compute $\gamma_i = \frac{t_i}{2u} + \frac{1-t_i}{2(1-u)}$ for $i = 1, ..., m$

7   **for** *e in [0, E]* **do**

8      Sample mini-batch data $\mathcal{B}$ from $\mathcal{D}$

9      Compute the gradients of the KL divergence:

$$g_1 = \nabla_\theta \log \frac{p(w)}{p(w|\theta)}$$

10      Compute the gradients of the IPM term with reparameterization trick:

$$g_2 = \nabla_\theta \alpha \mathrm{IPM}_G(\hat{p}_\Phi^{t=1}, \hat{p}_\Phi^{t=0})$$

11      Compute the gradients of the empirical loss with reparameterization trick:

$$g_3 = \nabla_\theta \frac{1}{|\mathcal{B}|} \sum_{i=1}^{|\mathcal{B}|} w_i L(y_i, f(\Phi(x_i), t_i))$$

12      Obtain the step size scalar $\eta$ with the Adam

13      Update the parameters:

$$W \leftarrow W - \eta(g_1 + g_2 + g_3)$$

14   **end**

---

Table 3: Statistics of the datasets used in our experiments.

| Dataset | #Replications | #Units | #Covariates | Treated Ratio | Control Ratio |
|---|---|---|---|---|---|
| ACIC | 100 | 4,802 | 82 | 28% | 72% |
| IHDP | 1,000 | 747 | 25 | 19% | 81% |
| Sim | 100 | 10,000 | 50 | 50% | 50% |
| ITE_kuaishou | 1 | 2,774,786 | 13 | 20% | 80% |

