# OpenReview forum: "Weight Uncertainty in Individual Treatment Effect"
_ICLR.cc/2024/Conference — ICLR 2024 Conference Withdrawn Submission_

### Official Review · Reviewer_GUVX · 2023-10-24

**Soundness:** 2 fair
**Presentation:** 1 poor
**Contribution:** 1 poor
**Rating:** 3
**Confidence:** 4

**Summary:**

The paper extends the balanced representation learning framework for CATE estimation from Shalit et al. (2017) to a Bayesian framework. For this purpose, the main result from Shalit et al. is adapted and an IPM penalty is added to the ELBO variational objective. Various experiments show competitive performance against existing CATE estimators.

**Strengths:**

- The paper is the first to provide a Bayesian framework for learning balanced CATE representations
- The proposed method is benchmarked against numerous baselines and performs competitively

**Weaknesses:**

- Novelty: First of all, the paper is **not** the first to address Bayesian inference for CATE estimation/ uncertainty quantification: Already Jesson et al. (2020) showed that a Bayesian approach can quantify uncertainty due to lack of data/overlap in CATE estimation. Hence the contribution of the paper is to combine Bayesian inference for CATE with balanced representations, which in my opinion is a rather straightforward thing to do. The paper heavily builds upon the work of Shalit et al. (2017) who originally proposed the balanced representation framework. From my understanding, the only novel result is the Bayesian version of the original generalization bound (Theorem 1/2), which again builds heavily on the original bound proposed by Shalit et al.

- Uncertainty quantification: From my understanding, the main benefit of using a Bayesian framework (and also the claim of the paper) is to allow for uncertainty quantification. However, in the experiments, the authors only evaluate performance in terms of point prediction. The "uncertainty study" provided evaluates some kind of robustness properties of the baselines, but **not** uncertainty for CATE estimation. I would have liked to see some experiments that evaluate the actual posterior CATE distribution, potentially showing benefits for decision-making as in Jesson et al (2020/2021).

- Implementation: Some implementation details are not quite clear to me. From my understanding, the authors add the IPM penalty to the ELBO loss (Eq. 12). Here, it seems like the likelihood B from Eq. 11 is replaced with an L2 loss, which would imply that the likelihood is modeled as a Gaussian with learnable mean and constant variance. I am not sure if this is correct, as the likelihood was never properly defined in the paper. This also would render the method incapable of useful uncertainty quantification for heteroscedastic data. Finally, the details regarding hyperparameters of the baselines are not reported and no hyperparameter tuning was performed, which raises concerns regarding the validity of the experimental results.

- Related to the previous point, **no Code is provided**, which makes it impossible to check the implementations of the proposed methods or the baselines.

- I am aware that Shalit et al. call the estimand of interest individual treatment effect (ITE). However, the majority of causal inference literature agrees on the naming "conditional average treatment effect" (CATE) or "heterogenous treatment effect" (HTE), as the effect is not really "individual" (two different individuals with the same covariates have equal CATE). ITE is usually reserved for the (random) difference in potential outcomes $Y_1 - Y_0$. I would suggest changing the naming from ITE to CATE to be consistent with the literature.

- Typos/Grammer/Notation: There is a substantial amount of typos, grammar issues, and notational inconsistencies in the paper, starting already with the **title** (Weight uncertainty in individual treatment effect). Some other examples: Appendix A.1 "proofs in ITE"; p.12 (top)", we have the following results"; Algorithm 1: $e \in [0, E]$ (looks like continuous epochs).

- Sec. 3 seems to provide (standard) background on Bayesian variational inference, not specific to the CATE setting. I suggest moving this section to the Appendix if space becomes an issue.

**Questions:**

- Can you provide more details/motivation regarding your implementation details? How is the likelihood defined? Why was only a simple normal distribution chosen as a variation family for the posterior? Please also provide all missing implementation details, including the code.

- Can you assess the proposed method in terms of uncertainty quantification? Also, a comparison with previously proposed Bayesian CATE methods (Jesson et al 2020) would be nice.

---

### Official Review · Reviewer_dZmK · 2023-10-28

**Soundness:** 2 fair
**Presentation:** 2 fair
**Contribution:** 2 fair
**Rating:** 5
**Confidence:** 3

**Summary:**

This paper proposes a framework to account for estimation uncertainty in estimating individual treatment effects (ITE). Under the standard consistency, overlap, and ignorability conditions, the authors leverage a previous result on estimation error of ITEs to establish a PAC-bayes bound on the estimation error, which inspires a Bayesian approach for estimation. Experiments on synthetic and real datasets show the performance of the proposed methods compared with existing competitors.

**Strengths:**

1. This paper studies a significant problem: ITE estimation, and it is sensible to account for estimation uncertainty.

2. The theoretical results are relatively intuitive to comprehend.

**Weaknesses:**

1. The presentation could be improved to further justify the proposed methods. For instance, the transition from PAC-bayes bound to Bayesian inference is not well-grounded. Why considering the intergrated posterior loss is helpful for one specific instance? It is also unclear to me how we can go from Bayesian inference to an ITE estimator.

2. The experimental results are not super strong. For instance, the proposed methods are less powerful than some competitors in many cases. If that is not improvable, more discussion should be provided for understanding in what cases the proposed methods may perform better.

More specific comments are in the Questions section.

**Questions:**

- What's the definition of "weights"?

In Section 3, the name "weights" $w$ appear out of nowhere and I'm not sure what it stands for. Also I don't understand what $\mathbb{E}_{p(w|D)[p(\hat{y}|x,t)]$ means, as $w$ doesn't appear in the quantity being integrated.

- What do you refer to as "distance metric" in this sentence "We refer to the model minimizing equation 12 with WASS distance metric as UITEWASS and the variant with MMD distance metric as UITEMMD."?

- In Section 7, UITE cannot beat existing methods for ACIC (witha clear margin). Is there any intuition on why? It can be helpful for understanding in what cases the proposed method could perform well.

- Why is NeuMFu sensitive to learning rate in Figure 2?

- How to tune the parameter $\alpha$ which is discussed in Section 7.5?

-------
Typos and wording issues:

1. The integral in equation (8) seems incomplete.

2. equation (9) should have a period.

---

### Official Review · Reviewer_A9su · 2023-10-31

**Soundness:** 2 fair
**Presentation:** 1 poor
**Contribution:** 2 fair
**Rating:** 3
**Confidence:** 3

**Summary:**

This paper studies the estimation of ITE (CATE) from a Bayesian perspective. Based on a variational bound on the generalization error, the proposed method obtains an ITE estimator via optimizing a regularized loss function. The method is evaluated on synthetic datasets, and through real data experiments.

**Strengths:**

It is interesting to see the Bayesian perspective leads to a regularized optimization problem,
and the resulting ITE estimator achieves good empirical performance.

**Weaknesses:**

The presentation of this paper is confusing and  can be improved. Please see my detailed comments in
the question section.

**Questions:**

1. I might have missed something, but I am confused what "uncertainty" refers to throughout the paper. Is it the uncertainty from the randomness of the
samples, or the prior, or anything else. It is also not clear to me what "robust" means in this context --- is it the robustness to distributional shift, or the
robustness in different environments, etc. I guess my main confusion is from understanding how the new method is "incorporating the uncertainty", and why it is  "providing robust ITE estimation".
It might be helpful to explicitly explain what these words mean as they appear multiple times in the paper.

2. I am also confused by the two losses in definition 1. Consider the simplest case, where there are no covariates.
The factual loss is $P(T=1)\int L(y,f(1))p(y|1)dy  + P(T=0)\int L(y,f(0))p(y|0)dy $,
and the counterfactual loss is $P(T=0)\int L(y,f(0))p(y|0)dy  + P(T=1)\int L(y,f(1))p(y|1)dy$ ---
aren't they the same thing? Would the $f(\Phi(x),1-t)$ be, say,  $f(\Phi(x),t)$  in the CF loss?


3. On page 3, the beginning of Section 3, are $\hat{Y}$ or $w$ defined?

4. Theorem 2 establishes an upper bound for $E_{f,\Phi\sim \rho}\epsilon_{\text{PEHE}}(f,\Phi)$.
It is a bit hand-wavy to me why optimizing the objective function obtained from the upper bound
would lead to a more "robust" model. More rigorous arguments are needed.

---

### Official Review · Reviewer_AZrT · 2023-10-31

**Soundness:** 2 fair
**Presentation:** 3 good
**Contribution:** 1 poor
**Rating:** 1
**Confidence:** 4

**Summary:**

The paper proposes a new method called UITE to estimate individual treatment effects (ITE) by modeling uncertainty through Bayesian inference, demonstrating decent performance over baseline approaches on a few datasets.

**Strengths:**

- Decently well written. Uncertainty estimation for TEE is interesting problem.
- Haven't seen the weight uncertainty approach by Blundell applied in causal inference.
- Good performance on select datasets on selected settings.

**Weaknesses:**

- Seems like the paper missed the entire body of literature on uncertainty in TEE. Use of GPs, Bayesian Neural Networks and Approximate Bayesian Methods like MC Dropout. No comparisons, no related work.
- Not sure if the correct evaluations are being used to show the benefit of uncertainty estimation (good calibration etc.) and if the method has good uncertainty estimation. This can be done on all the datasets (not just recommender system).
- The experimental results are not compelling. Not great performance and none the baselines I would expect to see.

**Questions:**

Above.

---

### Official Review · Reviewer_iMow · 2023-11-02

**Soundness:** 2 fair
**Presentation:** 3 good
**Contribution:** 2 fair
**Rating:** 5
**Confidence:** 3

**Summary:**

The paper introduces a new method for estimating ITE by leveraging a new objective function which connects with Bayesian inference. The paper demostrates the resulting algorithm in both simulated examples, semi-synthetic examples and an industry example.

**Strengths:**

Originality: The idea of using (12) as an objective in estimating ITE is new.

Quality: The experiments are comprehensive and the connection to previous literature is clear. The replication materials are sound and complete.

Clarity: The paper is easy to follow.

**Weaknesses:**

Less significant weakness: there are some typos in the text: e.g. (14) on page 7 and start of some words are unnessarily capitalized.
More significant weakness:
The experiements are already pretty comprehensive, however, I want to see how robust the results are to your model architecture choices.
The results are not convincing enough to say the proposed method outperforms in most cases. After all there are only 3 examples in table 2.

**Questions:**

Maybe I miss something, can you elaborate on why your method has much smaller standard errors, especially in IHDP?